# Dynamic Clustering via Asymptotics of the Dependent Dirichlet Process Mixture

**Trevor Campbell**
MIT
Cambridge, MA 02139
tdjc@mit.edu

**Miao Liu**
Duke University
Durham, NC 27708
miao.liu@duke.edu

**Brian Kulis**
Ohio State University
Columbus, OH 43210
kulis@cse.ohio-state.edu

**Jonathan P. How**
MIT
Cambridge, MA 02139
jhow@mit.edu

**Lawrence Carin**
Duke University
Durham, NC 27708
lcarin@duke.edu

## Abstract

This paper presents a novel algorithm, based upon the dependent Dirichlet process mixture model (DDPMM), for clustering batch-sequential data containing an unknown number of evolving clusters. The algorithm is derived via a low-variance asymptotic analysis of the Gibbs sampling algorithm for the DDPMM, and provides a hard clustering with convergence guarantees similar to those of the k-means algorithm. Empirical results from a synthetic test with moving Gaussian clusters and a test with real ADS-B aircraft trajectory data demonstrate that the algorithm requires orders of magnitude less computational time than contemporary probabilistic and hard clustering algorithms, while providing higher accuracy on the examined datasets.

## 1 Introduction

The Dirichlet process mixture model (DPMM) is a powerful tool for clustering data that enables the inference of an unbounded number of mixture components, and has been widely studied in the machine learning and statistics communities [1–4]. Despite its flexibility, it assumes the observations are exchangeable, and therefore that the data points have no inherent ordering that influences their labeling. This assumption is invalid for modeling temporally/spatially evolving phenomena, in which the order of the data points plays a principal role in creating meaningful clusters. The dependent Dirichlet process (DDP), originally formulated by MacEachern [5], provides a prior over such evolving mixture models, and is a promising tool for incrementally monitoring the dynamic evolution of the cluster structure within a dataset. More recently, a construction of the DDP built upon completely random measures [6] led to the development of the dependent Dirichlet process Mixture model (DDPMM) and a corresponding approximate posterior inference Gibbs sampling algorithm. This model generalizes the DPMM by including birth, death and transition processes for the clusters in the model.

The DDPMM is a Bayesian nonparametric (BNP) model, part of an ever-growing class of probabilistic models for which inference captures uncertainty in both the number of parameters and their values. While these models are powerful in their capability to capture complex structures in data without requiring explicit model selection, they suffer some practical shortcomings. Inference techniques for BNPs typically fall into two classes: sampling methods (e.g., Gibbs sampling [2]

or particle learning [4]) and optimization methods (e.g., variational inference [3] or stochastic variational inference [7]). Current methods based on sampling do not scale well with the size of the dataset [8]. Most optimization methods require analytic derivatives and the selection of an upper bound on the number of clusters a priori, where the computational complexity increases with that upper bound [3, 7]. State-of-the-art techniques in both classes are not ideal for use in contexts where performing inference quickly and reliably on large volumes of streaming data is crucial for timely decision-making, such as autonomous robotic systems [9–11]. On the other hand, many classical clustering methods [12–14] scale well with the size of the dataset and are easy to implement, and advances have recently been made to capture the flexibility of Bayesian nonparametrics in such approaches [15]. However, as of yet there is no classical algorithm that captures dynamic cluster structure with the same representational power as the DDP mixture model.

This paper discusses the Dynamic Means algorithm, a novel hard clustering algorithm for spatio-temporal data derived from the low-variance asymptotic limit of the Gibbs sampling algorithm for the dependent Dirichlet process Gaussian mixture model. This algorithm captures the scalability and ease of implementation of classical clustering methods, along with the representational power of the DDP prior, and is guaranteed to converge to a local minimum of a k-means-like cost function. The algorithm is significantly more computationally tractable than Gibbs sampling, particle learning, and variational inference for the DDP mixture model in practice, while providing equivalent or better clustering accuracy on the examples presented. The performance and characteristics of the algorithm are demonstrated in a test on synthetic data, with a comparison to those of Gibbs sampling, particle learning and variational inference. Finally, the applicability of the algorithm to real data is presented through an example of clustering a spatio-temporal dataset of aircraft trajectories recorded across the United States.

## 2 Background

The Dirichlet process (DP) is a prior over mixture models, where the number of mixture components is not known a priori[16]. In general, we denote $D \sim \text{DP}(\mu)$, where $\alpha_\mu \in \mathbb{R}_+$ and $\mu : \Omega \to \mathbb{R}_+, \int_\Omega d\mu = \alpha_\mu$ are the *concentration parameter* and *base measure* of the DP, respectively. If $D \sim \text{DP}$, then $D = \{(\theta_k, \pi_k)\}_{k=0}^\infty \subset \Omega \times \mathbb{R}_+$, where $\theta_k \in \Omega$ and $\pi_k \in \mathbb{R}_+$[17]. The reader is directed to [1] for a more thorough coverage of Dirichlet processes.

The dependent Dirichlet process (DDP)[5], an extension to the DP, is a prior over evolving mixture models. Given a Poisson process construction[6], the DDP essentially forms a Markov chain of DPs $(D_1, D_2, \dots)$, where the transitions are governed by a set of three stochastic operations: Points $\theta_k$ may be added, removed, and may move during each step of the Markov chain. Thus, they become parameterized by time, denoted by $\theta_{kt}$. In slightly more detail, if $D_t$ is the DP at time step $t$, then the following procedure defines the generative model of $D_t$ conditioned on $D_{t-1} \sim \text{DP}(\mu_{t-1})$:

1. **Subsampling**: Define a function $q : \Omega \to [0, 1]$. Then for each point $(\theta, \pi) \in D_{t-1}$, sample a Bernoulli distribution $b_\theta \sim \text{Be}(q(\theta))$. Set $D'_t$ to be the collection of points $(\theta, \pi)$ such that $b_\theta = 1$, and renormalize the weights. Then $D'_t \sim \text{DP}(q\mu_{t-1})$, where $(q\mu)(A) = \int_A q(\theta)\mu(d\theta)$.

2. **Transition**: Define a distribution $T : \Omega \times \Omega \to \mathbb{R}_+$. For each point $(\theta, \pi) \in D'_t$, sample $\theta' \sim T(\theta'|\theta)$, and set $D''_t$ to be the collection of points $(\theta', \pi)$. Then $D''_t \sim \text{DP}(Tq\mu_{t-1})$, where $(T\mu)(A) = \int_A \int_\Omega T(\theta'|\theta)\mu(d\theta)$.

3. **Superposition**: Sample $F \sim \text{DP}(\nu)$, and sample $(c_D, c_F) \sim \text{Dir}(Tq\mu_{t-1}(\Omega), \nu(\Omega))$. Then set $D_t$ to be the union of $(\theta, c_D\pi)$ for all $(\theta, \pi) \in D''_t$ and $(\theta, c_F\pi)$ for all $(\theta, \pi) \in F$. Thus, $D_t$ is a random convex combination of $D''_t$ and $F$, where $D_t \sim \text{DP}(Tq\mu_{t-1} + \nu)$.

If the DDP is used as a prior over a mixture model, these three operations allow new mixture components to arise over time, and old mixture components to exhibit dynamics and perhaps disappear over time. As this is covered thoroughly in [6], the mathematics of the underlying Poisson point process construction are not discussed in more depth in this work. However, an important result of using such a construction is the development of an explicit posterior for $D_t$ given observations of the points $\theta_{kt}$ at timestep $t$. For each point $k$ that was observed in $D_\tau$ for some $\tau : 1 \le \tau \le t$, define: $n_{kt} \in \mathbb{N}$ as the number of observations of point $k$ in timestep $t$; $c_{kt} \in \mathbb{N}$ as the number of past

observations of point $k$ prior to timestep $t$, i.e. $c_{kt} = \sum_{\tau=1}^{t-1} n_{k\tau}$; $q_{kt} \in (0,1)$ as the subsampling weight on point $k$ at timestep $t$; and $\Delta t_k$ as the number of time steps that have elapsed since point $k$ was last observed. Further, let $\nu_t$ be the measure for unobserved points at time step $t$. Then,

$$D_t | D_{t-1} \sim \text{DP}\left(\nu_t + \sum_{k:n_{kt}=0} q_{kt} c_{kt} T(\cdot\,|\theta_{k(t-\Delta t_k)}) + \sum_{k:n_{kt}>0} (c_{kt} + n_{kt})\delta_{\theta_{kt}}\right) \qquad (1)$$

where $c_{kt} = 0$ for any point $k$ that was first observed during timestep $t$. This posterior leads directly to the development of a Gibbs sampling algorithm for the DDP, whose low-variance asymptotics are discussed further below.

## 3 Asymptotic Analysis of the DDP Mixture

The dependent Dirichlet process Gaussian mixture model (DDP-GMM) serves as the foundation upon which the present work is built. The generative model of a DDP-GMM at time step $t$ is

$$\{\theta_{kt}, \pi_{kt}\}_{k=1}^{\infty} \sim \text{DP}(\mu_t)$$
$$\{z_{it}\}_{i=1}^{N_t} \sim \text{Categorical}(\{\pi_{kt}\}_{k=1}^{\infty}) \qquad (2)$$
$$\{y_{it}\}_{i=1}^{N_t} \sim \mathcal{N}(\theta_{z_{it}t}, \sigma I)$$

where $\theta_{kt}$ is the mean of cluster $k$, $\pi_{kt}$ is the categorical weight for class $k$, $y_{it}$ is a $d$-dimensional observation vector, $z_{it}$ is a cluster label for observation $i$, and $\mu_t$ is the base measure from equation (1). Throughout the rest of this paper, the subscript $kt$ refers to quantities related to cluster $k$ at time step $t$, and subscript $it$ refers to quantities related to observation $i$ at time step $t$.

The Gibbs sampling algorithm for the DDP-GMM iterates between sampling labels $z_{it}$ for datapoints $y_{it}$ given the set of parameters $\{\theta_{kt}\}$, and sampling parameters $\theta_{kt}$ given each group of data $\{y_{it} : z_{it} = k\}$. Assuming the transition model $T$ is Gaussian, and the subsampling function $q$ is constant, the functions and distributions used in the Gibbs sampling algorithm are: the prior over cluster parameters, $\theta \sim \mathcal{N}(\phi, \rho I)$; the likelihood of an observation given its cluster parameter, $y_{it} \sim \mathcal{N}(\theta_{kt}, \sigma I)$; the distribution over the transitioned cluster parameter given its last known location after $\Delta t_k$ time steps, $\theta_{kt} \sim \mathcal{N}(\theta_{k(t-\Delta t_k)}, \xi\Delta t_k I)$; and the subsampling function $q(\theta) = q \in (0,1)$. Given these functions and distributions, the low-variance asymptotic limits (i.e. $\sigma \to 0$) of these two steps are discussed in the following sections.

### 3.1 Setting Labels Given Parameters

In the label sampling step, a datapoint $y_{it}$ can either create a new cluster, join a current cluster, or revive an old, transitioned cluster. Using the distributions defined previously, the label assignment probabilities are

$$p(z_{it} = k | \dots) \propto \begin{cases} \alpha_t (2\pi(\sigma + \rho))^{-d/2} \exp\left(-\frac{||y_{it} - \phi||^2}{2(\sigma + \rho)}\right) & k = K+1 \\ (c_{kt} + n_{kt})(2\pi\sigma)^{-d/2} \exp\left(-\frac{||y_{it} - \theta_{kt}||^2}{2\sigma}\right) & n_{kt} > 0 \\ q_{kt} c_{kt} (2\pi(\sigma + \xi\Delta t_k))^{-d/2} \exp\left(-\frac{||y_{it} - \theta_{k(t-\Delta t_k)}||^2}{2(\sigma + \xi\Delta t_k)}\right) & n_{kt} = 0 \end{cases} \qquad (3)$$

where $q_{kt} = q^{\Delta t_k}$ due to the fact that $q(\theta)$ is constant over $\Omega$, and $\alpha_t = \alpha_\nu \frac{1-q^t}{1-q}$ where $\alpha_\nu$ is the concentration parameter for the innovation process, $F_t$. The low-variance asymptotic limit of this label assignment step yields meaningful assignments as long as $\alpha_\nu$, $\xi$, and $q$ vary appropriately with $\sigma$; thus, setting $\alpha_\nu$, $\xi$, and $q$ as follows (where $\lambda$, $\tau$ and $Q$ are positive constants):

$$\alpha_\nu = (1 + \rho/\sigma)^{d/2} \exp\left(-\frac{\lambda}{2\sigma}\right), \quad \xi = \tau\sigma, \quad q = \exp\left(-\frac{Q}{2\sigma}\right) \qquad (4)$$

yields the following assignments in the limit as $\sigma \to 0$:

$$z_{it} = \arg\min_k \{J_k\}, \; J_k = \begin{cases} ||y_{it} - \theta_{kt}||^2 & \text{if } \theta_k \text{ instantiated} \\ Q\Delta t_k + \frac{||y_{it} - \theta_{k(t-\Delta t_k)}||^2}{\tau\Delta t_k + 1} & \text{if } \theta_k \text{ old, uninstantiated} \\ \lambda & \text{if } \theta_k \text{ new} \end{cases} \cdot \qquad (5)$$

In this assignment step, $Q\Delta t_k$ acts as a cost penalty for reviving old clusters that increases with the time since the cluster was last seen, $\tau\Delta t_k$ acts as a cost reduction to account for the possible motion of clusters since they were last instantiated, and $\lambda$ acts as a cost penalty for introducing a new cluster.

## 3.2 Setting Parameters given Labels

In the parameter sampling step, the parameters are sampled using the distribution

$$\mathrm{p}(\theta_{kt}|\{y_{it} : z_{it} = k\}) \propto p(\{y_{it} : z_{it} = k\}|\theta_{kt})\mathrm{p}(\theta_{kt}) \tag{6}$$

There are two cases to consider when setting a parameter $\theta_{kt}$. Either $\Delta t_k = 0$ and the cluster is new in the current time step, or $\Delta t_k > 0$ and the cluster was previously created, disappeared for some amount of time, and then was revived in the current time step.

**New Cluster**  Suppose cluster $k$ is being newly created. In this case, $\theta_{kt} \sim \mathcal{N}(\phi, \rho)$. Using the fact that a normal prior is conjugate a normal likelihood, the closed-form posterior for $\theta_{kt}$ is

$$\theta_{kt}|\{y_{it} : z_{it} = k\} \sim \mathcal{N}\left(\theta_{\text{post}}, \sigma_{\text{post}}\right)$$

$$\theta_{\text{post}} = \sigma_{\text{post}}\left(\frac{\phi}{\rho} + \frac{\sum_{i=1}^{n_{kt}} y_{it}}{\sigma}\right), \ \sigma_{\text{post}} = \left(\frac{1}{\rho} + \frac{n_{kt}}{\sigma}\right)^{-1} \tag{7}$$

Then letting $\sigma \to 0$,

$$\theta_{kt} = \frac{\left(\sum_{i=1}^{n_{kt}} y_{it}\right)}{n_{kt}} \overset{\text{def}}{=} m_{kt} \tag{8}$$

where $m_{kt}$ is the mean of the observations in the current timestep.

**Revived Cluster**  Suppose there are $\Delta t_k$ time steps where cluster $k$ was not observed, but there are now $n_{kt}$ data points with mean $m_{kt}$ assigned to it in this time step. In this case,

$$\mathrm{p}(\theta_{kt}) = \int_{\theta} T(\theta_{kt}|\theta)\mathrm{p}(\theta)\,\mathrm{d}\theta, \ \theta \sim \mathcal{N}(\theta', \sigma'). \tag{9}$$

Again using conjugacy of normal likelihoods and priors,

$$\theta_{kt}|\{y_{it} : z_{it} = k\} \sim \mathcal{N}\left(\theta_{\text{post}}, \sigma_{\text{post}}\right)$$

$$\theta_{\text{post}} = \sigma_{\text{post}}\left(\frac{\theta'}{\xi\Delta t_k + \sigma'} + \frac{\sum_{i=1}^{n_{kt}} y_{it}}{\sigma}\right), \ \sigma_{\text{post}} = \left(\frac{1}{\xi\Delta t_k + \sigma'} + \frac{n_{kt}}{\sigma}\right)^{-1} \tag{10}$$

Similarly to the label assignment step, let $\xi = \tau\sigma$. Then as long as $\sigma' = \sigma/w$, $w > 0$ (which holds if equation (10) is used to recursively keep track of the parameter posterior), taking the asymptotic limit of this as $\sigma \to 0$ yields:

$$\theta_{kt} = \frac{\theta'(w^{-1} + \Delta t_k\tau)^{-1} + n_{kt}m_{kt}}{(w^{-1} + \Delta t_k\tau)^{-1} + n_{kt}} \tag{11}$$

that is to say, the revived $\theta_{kt}$ is a weighted average of estimates using current timestep data and previous timestep data. $\tau$ controls how much the current data is favored - as $\tau$ increases, the weight on current data increases, which is explained by the fact that our uncertainty in where the old $\theta'$ transitioned to increases with $\tau$. It is also noted that if $\tau = 0$, this reduces to a simple weighted average using the amount of data collected as weights.

**Combined Update**  Combining the updates for new cluster parameters and old transitioned cluster parameters yields a recursive update scheme:

$$
\begin{aligned}
\gamma_{kt} &= \left((w_{k(t-\Delta t_k)})^{-1} + \Delta t_k\tau\right)^{-1} \\
\theta_{k0} = m_{k0} \quad & \text{and} \quad \theta_{kt} = \frac{\theta_{k(t-\Delta t_k)}\gamma_{kt} + n_{kt}m_{kt}}{\gamma_{kt} + n_{kt}} \\
w_{k0} = n_{k0} & \\
& w_{kt} = \gamma_{kt} + n_{kt}
\end{aligned}
\tag{12}
$$

where time step 0 here corresponds to when the cluster is first created. An interesting interpretation of this update is that it behaves like a standard Kalman filter, in which $w_{kt}^{-1}$ serves as the current estimate variance, $\tau$ serves as the process noise variance, and $n_{kt}$ serves as the inverse of the measurement variance.

| **Algorithm 1** Dynamic Means | **Algorithm 2** CLUSTER |
|---|---|
| **Input:** $\{\mathcal{Y}_t\}_{t=1}^{t_f}, Q, \lambda, \tau$ <br> $\quad \mathcal{C}_1 \leftarrow \emptyset$ <br> $\quad$ **for** $t = 1 \rightarrow t_f$ **do** <br> $\quad\quad (\mathcal{K}_t, \mathcal{Z}_t, L_t) \leftarrow$ CLUSTER$(\mathcal{Y}_t, \mathcal{C}_t, Q, \lambda, \tau)$ <br> $\quad\quad \mathcal{C}_{t+1} \leftarrow$ UPDATEC$(\mathcal{Z}_t, \mathcal{K}_t, \mathcal{C}_t)$ <br> $\quad$ **end for** <br> $\quad$ **return** $\{\mathcal{K}_t, \mathcal{Z}_t, L_t\}_{t=1}^{t_f}$ | **Input:** $\mathcal{Y}_t, \mathcal{C}_t, Q, \lambda, \tau$ <br> $\quad \mathcal{K}_t \leftarrow \emptyset, \mathcal{Z}_t \leftarrow \emptyset, L_0 \leftarrow \infty$ <br> $\quad$ **for** $n = 1 \rightarrow \infty$ **do** <br> $\quad\quad (\mathcal{Z}_t, \mathcal{K}_t) \leftarrow$ ASSIGNLABELS$(\mathcal{Y}_t, \mathcal{Z}_t, \mathcal{K}_t, \mathcal{C}_t)$ <br> $\quad\quad (\mathcal{K}_t, L_n) \leftarrow$ ASSIGNPARAMS$(\mathcal{Y}_t, \mathcal{Z}_t, \mathcal{C}_t)$ <br> $\quad\quad$ **if** $L_n = L_{n-1}$ **then** <br> $\quad\quad\quad$ **return** $\mathcal{K}_t, \mathcal{Z}_t, L_n$ <br> $\quad\quad$ **end if** <br> $\quad$ **end for** |

## 4 The Dynamic Means Algorithm

In this section, some further notation is required for brevity:

$$\begin{aligned} \mathcal{Y}_t &= \{y_{it}\}_{i=1}^{N_t}, \quad \mathcal{Z}_t = \{z_{it}\}_{i=1}^{N_t} \\ \mathcal{K}_t &= \{(\theta_{kt}, w_{kt}) : n_{kt} > 0\}, \quad \mathcal{C}_t = \{(\Delta t_k, \theta_{k(t-\Delta t_k)}, w_{k(t-\Delta t_k)})\} \end{aligned} \quad (13)$$

where $\mathcal{Y}_t$ and $\mathcal{Z}_t$ are the sets of observations and labels at time step $t$, $\mathcal{K}_t$ is the set of currently active clusters (some are new with $\Delta t_k = 0$, and some are revived with $\Delta t_k > 0$), and $\mathcal{C}_t$ is the set of old cluster information.

### 4.1 Algorithm Description

As shown in the previous section, the low-variance asymptotic limit of the DDP Gibbs sampling algorithm is a deterministic observation label update (5) followed by a deterministic, weighted least-squares parameter update (12). Inspired by the original K-Means algorithm, applying these two updates iteratively yields an algorithm which clusters a set of observations at a single time step given cluster means and weights from past time steps (Algorithm 2). Applying Algorithm 2 to a sequence of batches of data yields a clustering procedure that is able to track a set of dynamically evolving clusters (Algorithm 1), and allows new clusters to emerge and old clusters to be forgotten. While this is the primary application of Algorithm 1, the sequence of batches need not be a temporal sequence. For example, Algorithm 1 may be used as an any-time clustering algorithm for large datasets, where the sequence of batches is generated by selecting random subsets of the full dataset.

The ASSIGNPARAMS function is exactly the update from equation (12) applied to each $k \in \mathcal{K}_t$. Similarly, the ASSIGNLABELS function applies the update from equation (5) to each observation; however, in the case that a new cluster is created or an old one is revived by an observation, ASSIGNLABELS also creates a parameter for that new cluster based on the parameter update equation (12) with that single observation. Note that the performance of the algorithm depends on the order in which ASSIGNLABELS assigns labels. Multiple random restarts of the algorithm with different assignment orders may be used to mitigate this dependence. The UPDATEC function is run after clustering observations from each time step, and constructs $\mathcal{C}_{t+1}$ by setting $\Delta t_k = 1$ for any new or revived cluster, and by incrementing $\Delta t_k$ for any old cluster that was not revived:

$$\mathcal{C}_{t+1} = \{(\Delta t_k + 1, \theta_{k(t-\Delta t_k)}, w_{k(t-\Delta t_k)}) : k \in \mathcal{C}_t, k \notin \mathcal{K}_t\} \cup \{(1, \theta_{kt}, w_{kt}) : k \in \mathcal{K}_t\} \quad (14)$$

An important question is whether this algorithm is guaranteed to converge while clustering data in each time step. Indeed, it is; Theorem 1 shows that a particular cost function $L_t$ monotonically decreases under the label and parameter updates (5) and (12) at each time step. Since $L_t \geq 0$, and it is monotonically decreased by Algorithm 2, the algorithm converges. Note that the Dynamic Means is only guaranteed to converge to a local optimum, similarly to the k-means[12] and DP-Means[15] algorithms.

**Theorem 1.** *Each iteration in Algorithm 2 monotonically decreases the cost function $L_t$, where*

$$L_t = \sum_{k \in \mathcal{K}_t} \left( \overbrace{\lambda\left[\Delta t_k = 0\right]}^{\text{New Cost}} + \overbrace{Q\Delta t_k}^{\text{Revival Cost}} + \overbrace{\gamma_{kt}||\theta_{kt} - \theta_{k(t-\Delta t_k)}||_2^2 + \sum_{\substack{y_{it} \in \mathcal{Y}_t \\ z_{it}=k}} ||y_{it} - \theta_{kt}||_2^2}^{\text{Weighted-Prior Sum-Squares Cost}} \right) \quad (15)$$

The cost function is comprised of a number of components for each currently active cluster $k \in \mathcal{K}_t$: A penalty for new clusters based on $\lambda$, a penalty for old clusters based on $Q$ and $\Delta t_k$, and finally

a prior-weighted sum of squared distance cost for all the observations in cluster $k$. It is noted that for new clusters, $\theta_{kt} = \theta_{k(t-\Delta t_k)}$ since $\Delta t_k = 0$, so the least squares cost is unweighted. The ASSIGNPARAMS function calculates this cost function in each iteration of Algorithm 2, and the algorithm terminates once the cost function does not decrease during an iteration.

## 4.2 Reparameterizing the Algorithm

In order to use the Dynamic Means algorithm, there are three free parameters to select: $\lambda$, $Q$, and $\tau$. While $\lambda$ represents how far an observation can be from a cluster before it is placed in a new cluster, and thus can be tuned intuitively, $Q$ and $\tau$ are not so straightforward. The parameter $Q$ represents a conceptual added distance from any data point to a cluster for every time step that the cluster is not observed. The parameter $\tau$ represents a conceptual reduction of distance from any data point to a cluster for every time step that the cluster is not observed. How these two quantities affect the algorithm, and how they interact with the setting of $\lambda$, is hard to judge.

Instead of picking $Q$ and $\tau$ directly, the algorithm may be reparameterized by picking $N_Q, k_\tau \in \mathbb{R}_+$, $N_Q > 1$, $k_\tau \geq 1$, and given a choice of $\lambda$, setting

$$Q = \lambda/N_Q \quad \tau = \frac{N_Q(k_\tau - 1) + 1}{N_Q - 1}. \tag{16}$$

If $Q$ and $\tau$ are set in this manner, $N_Q$ represents the number (possibly fractional) of time steps a cluster can be unobserved before the label update (5) will never revive that cluster, and $k_\tau \lambda$ represents the maximum squared distance away from a cluster center such that after a single time step, the label update (5) will revive that cluster. As $N_Q$ and $k_\tau$ are specified in terms of concrete algorithmic behavior, they are intuitively easier to set than $Q$ and $\tau$.

## 5 Related Work

Prior k-means clustering algorithms that determine the number of clusters present in the data have primarily involved a method for iteratively modifying k using various statistical criteria [13, 14, 18]. In contrast, this work derives this capability from a Bayesian nonparametric model, similarly to the DP-Means algorithm [15]. In this sense, the relationship between the Dynamic Means algorithm and the dependent Dirichlet process [6] is exactly that between the DP-Means algorithm and Dirichlet process [16], where the Dynamic Means algorithm may be seen as an extension to the DP-Means that handles sequential data with time-varying cluster parameters. MONIC [19] and MC3 [20] have the capability to monitor time-varying clusters; however, these methods require datapoints to be identifiable across timesteps, and determine cluster similarity across timesteps via the commonalities between label assignments. The Dynamic Means algorithm does not require such information, and tracks clusters essentially based on similarity of the parameters across timesteps. Evolutionary clustering [21, 22], similar to Dynamic Means, minimizes an objective consisting of a cost for clustering the present data set and a cost related to the comparison between the current clustering and past clusterings. The present work can be seen as a theoretically-founded extension of this class of algorithm that provides methods for automatic and adaptive prior weight selection, forming correspondences between old and current clusters, and for deciding when to introduce new clusters. Finally, some sequential Monte-Carlo methods (e.g. particle learning [23] or multi-target tracking [24, 25]) can be adapted for use in the present context, but suffer the drawbacks typical of particle filtering methods.

## 6 Applications

### 6.1 Synthetic Gaussian Motion Data

In this experiment, moving Gaussian clusters on $[0, 1] \times [0, 1]$ were generated synthetically over a period of 100 time steps. In each step, there was some number of clusters, each having 15 data points. The data points were sampled from a symmetric Gaussian distribution with a standard deviation of 0.05. Between time steps, the cluster centers moved randomly, with displacements sampled from the same distribution. At each time step, each cluster had a 0.05 probability of being destroyed.

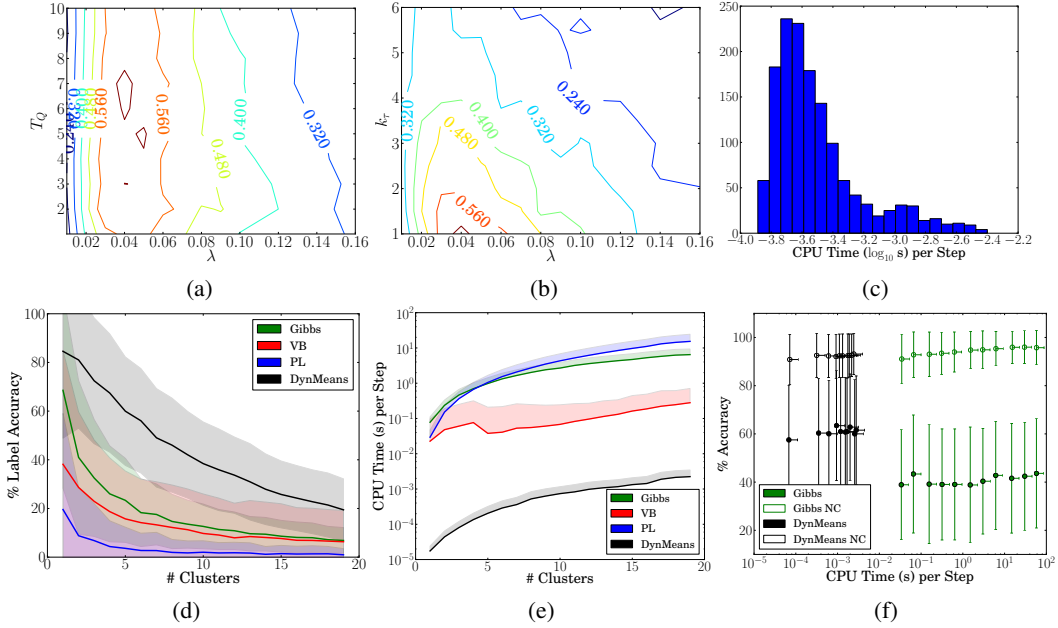

Figure 1: (1a - 1c): Accuracy contours and CPU time histogram for the Dynamic Means algorithm. (1d - 1e): Comparison with Gibbs sampling, variational inference, and particle learning. Shaded region indicates $1\sigma$ interval; in (1e), only upper half is shown. (1f): Comparison of accuracy when enforcing (Gibbs, DynMeans) and not enforcing (Gibbs NC, DynMeans NC) correct cluster tracking.

This data was clustered with Dynamic Means (with 3 random assignment ordering restarts), DDP-GMM Gibbs sampling [6], variational inference [3], and particle learning [4] on a computer with an Intel i7 processor and 16GB of memory. First, the number of clusters was fixed to 5, and the parameter space of each algorithm was searched for the best possible cluster label accuracy (taking into account correct cluster tracking across time steps). The results of this parameter sweep for the Dynamic Means algorithm with 50 trials at each parameter setting are shown in Figures 1a–1c. Figures 1a and 1b show how the average clustering accuracy varies with the parameters after fixing either $k_\tau$ or $T_Q$ to their values at the maximum accuracy parameter setting over the full space. The Dynamic Means algorithm had a similar robustness with respect to variations in its parameters as the comparison algorithms. The histogram in Figure 1c demonstrates that the clustering speed is robust to the setting of parameters. The speed of Dynamic Means, coupled with the smoothness of its performance with respect to its parameters, makes it well suited for automatic tuning [26].

Using the best parameter setting for each algorithm, the data as described above were clustered in 50 trials with a varying number of clusters present in the data. For the Dynamic Means algorithm, parameter values $\lambda = 0.04$, $T_Q = 6.8$, and $k_\tau = 1.01$ were used, and the algorithm was again given 3 attempts with random labeling assignment orders, where the lowest cost solution of the 3 was picked to proceed to the next time step. For the other algorithms, the parameter values $\alpha = 1$ and $q = 0.05$ were used, with a Gaussian transition distribution variance of 0.05. The number of samples for the Gibbs sampling algorithm was 5000 with one recorded for every 5 samples, the number of particles for the particle learning algorithm was 100, and the variational inference algorithm was run to a tolerance of $10^{-20}$ with the maximum number of iterations set to 5000.

In Figures 1d and 1e, the labeling accuracy and clustering time (respectively) for the algorithms is shown. The sampling algorithms were handicapped to generate Figure 1d; the best posterior sample in terms of labeling accuracy was selected at each time step, which required knowledge of the true labeling. Further, the accuracy computation included enforcing consistency across timesteps, to allow tracking individual cluster trajectories. If this is not enforced (i.e. accuracy considers each time step independently), the other algorithms provide accuracies more comparable to those of the Dynamic Means algorithm. This effect is demonstrated in Figure 1f, which shows the time/accuracy tradeoff for Gibbs sampling (varying the number of samples) and Dynamic Means (varying the number of restarts). These examples illustrate that Dynamic Means outperforms standard inference algorithms in both label accuracy and computation time for cluster tracking problems.

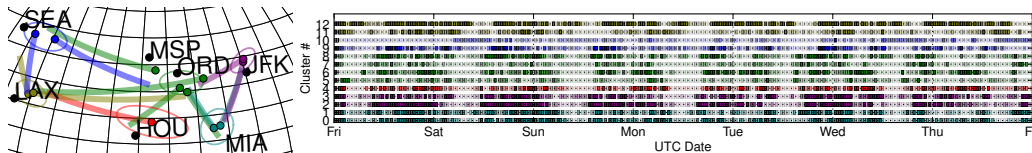

Figure 2: Results of the GP aircraft trajectory clustering. Left: A map (labeled with major US city airports) showing the overall aircraft flows for 12 trajectories, with colors and $1\sigma$ confidence ellipses corresponding to takeoff region (multiple clusters per takeoff region), colored dots indicating mean takeoff position for each cluster, and lines indicating the mean trajectory for each cluster. Right: A track of plane counts for the 12 clusters during the week, with color intensity proportional to the number of takeoffs at each time.

## 6.2 Aircraft Trajectory Clustering

In this experiment, the Dynamic Means algorithm was used to find the typical spatial and temporal patterns in the motions of commercial aircraft. Automatic dependent surveillance-broadcast (ADS-B) data, including plane identification, timestamp, latitude, longitude, heading and speed, was collected from all transmitting planes across the United States during the week from 2013-3-22 1:30:0 to 2013-3-28 12:0:0 UTC. Then, individual ADS-B messages were connected together based on their plane identification and timestamp to form trajectories, and erroneous trajectories were filtered based on reasonable spatial/temporal bounds, yielding 17,895 unique trajectories. Then, for each trajectory, a Gaussian process was trained using the latitude and longitude of each ADS-B point along the trajectory as the inputs and the North and East components of plane velocity at those points as the outputs. Next, the mean latitudinal and longitudinal velocities from the Gaussian process were queried for each point on a regular lattice across the USA (10 latitudes and 20 longitudes), and used to create a 400-dimensional feature vector for each trajectory. Of the resulting 17,895 feature vectors, 600 were hand-labeled (each label including a confidence weight in $[0, 1]$). The feature vectors were clustered using the DP-Means algorithm on the entire dataset in a single batch, and using Dynamic Means / DDPGMM Gibbs sampling (with 50 samples) with half-hour takeoff window batches.

The results of this exercise are provided in Figure 2 and Table 1. Figure 2 shows the spatial and temporal properties of the 12 most popular clusters discovered by Dynamic Means, demonstrating that the algorithm successfully identified major flows of commercial aircraft across the US. Table 1 corroborates these qualitative results with a quantitative comparison of the computation time and accuracy for the three algorithms tested over 20 trials. The

Table 1: Mean computational time & accuracy on hand-labeled aircraft trajectory data

| Alg. | % Acc. | Time (s) |
|---|---|---|
| DynM | 55.9 | $2.7 \times 10^2$ |
| DPM | 55.6 | $3.1 \times 10^3$ |
| Gibbs | 36.9 | $1.4 \times 10^4$ |

confidence-weighted accuracy was computed by taking the ratio between the sum of the weights for correctly labeled points and the sum of all weights. The DDPGMM Gibbs sampling algorithm was handicapped as described in the synthetic experiment section. Of the three algorithms, Dynamic Means provided the highest labeling accuracy, while requiring orders of magnitude less computation time than both DP-Means and DDPGMM Gibbs sampling.

## 7 Conclusion

This work developed a clustering algorithm for batch-sequential data containing temporally evolving clusters, derived from a low-variance asymptotic analysis of the Gibbs sampling algorithm for the dependent Dirichlet process mixture model. Synthetic and real data experiments demonstrated that the algorithm requires orders of magnitude less computational time than contemporary probabilistic and hard clustering algorithms, while providing higher accuracy on the examined datasets. The speed of inference coupled with the convergence guarantees provided yield an algorithm which is suitable for use in time-critical applications, such as online model-based autonomous planning systems.

**Acknowledgments**

This work was supported by NSF award IIS-1217433 and ONR MURI grant N000141110688.

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
