[Reviews · NeurIPS 2013]

Submitted by Assigned_Reviewer_3

Summary: The paper presents a hard clustering algorithm for clustering batch sequential continuous data. The algorithm is derived by performing a low variance asymptotic analysis of the Gibbs sampling algorithm for the dependent Dirichlet process Gaussian mixture model (DDPMM).
=====
This is a well written paper that is easy to follow. The work is technically sound and I only have a few minor concerns.

Originality: Moderate. The authors perform a small variance asymptotic analysis for the DDPMM Gibbs sampler and although a similar analysis has previously been performed on the Dirichlet process mixture Gibbs sampler, the model and algorithm considered here are sufficiently different.

Significance: The paper is interesting in two ways: 1)It demonstrates that it is possible to perform small variance analysis on models more sophisticated than Dirichlet process mixtures and 2) It develops an efficient hard clustering algorithm for time evolving data.
=====

Detailed Comments:
1) It appears that the analysis depends critically on the transition and observation distributions being Gaussian. Are extensions to models using discrete observation and transition distributions straightforward?
2) It is a little concerning that the small variance analysis is performed on a particular inference algorithm (Gibbs sampling) for the DDPMM. Can something be said about the model directly, independent of the inference algorithm?
3) Selection of the free parameters of the algorithm seems cumbersome, performing an exhaustive parameter search is often not feasible for large datasets. How were the parameters chosen for the aircraft trajectory data?
4) In the toy experiment, how were the Gibbs sampler and VB initialized? Particularly poor initializations, for e.g., all data being assigned to the same cluster, may partly be responsible for the large performance gaps. Were multiple runs of VB performed?
5) It is difficult to judge the aircraft trajectory clustering results from the visualization alone. Quantitative numbers would be useful. How about L2 error on held out trajectories? This error would obviously be minimized if each trajectory is its own cluster, but can still be useful in comparing two clusterings with comparable number of clusters.
6) How were the Gibbs sampling results unreliable?
Summary: Overall, this is a solid piece of work and a well written paper. There are some minor concerns with the experimental section.

Submitted by Assigned_Reviewer_5

Summary: The paper presents a deterministic inference algorithm for a clustering
problem for batch sequential data (data in each batch exist in clusters, and
clusters across batches are related as per some evolving structure). The Dependent
Dirichlet Process Gaussian mixture model (DDP-GMM) is a nonparametric Bayesian method
for dealing with such data. The paper uses the idea of doing low-variance asymptotics
on the Gibbs sampling update equations in a DDP-GMM model (a similar approach was
recently proposed by Kulis and Jordan for DP-GMM in their DP-means algorithm) and
derives a deterministic algorithm that is (1) more efficient to implement than Gibbs
sampling for DDP-GMM, (2) can learn the number of clusters just like DDP-GMM. Experimental
results on a synthetic and a real dataset show that the proposed algorithm runs fasters
than other inference methods for DDP-GMM such as Gibbs sampling, variational inference,
and sequential Monte Carlo.

Quality: The technical quality is rigorous and the details appear to be correct.

Clarity: The paper is well-written and the exposition is easy to follow.

Originality: The paper is based on recently proposed idea of using low-variance
asymptotics on the Gibbs sampling update equations in case of DP mixture models.

Significance: The paper addresses an important problem (scaling up nonparametric
Bayesian methods for large datasets).

Paper Strengths:

- The proposed algorithm is intuitive, seems easy to implement, and is scalable for
large datasets.

- Although the basic recipe is borrowed from the DP-means work of Kulis and Jordan
(taking the Gibbs sampling updates and applying the low-variance asymptotics), the
ideas in section 3.2 (parameter updates) are rather novel in this context.

Paper Weaknesses:

- The algorithm has 3 free parameters. For an unsupervised algorithm (with no option
of doing cross-validation), this is a bit worrying. Although the paper gives some
suggestions on how to choose the parameters, it is not clear how well they will work
in practice.

- There should have been a comparison with a simple variant of the DP-means algorithm
(Kulis and Jordan, 2012) for batch-sequential data (see the suggestion in the comments
below on how to do it).

- On real-data, there is no comparison with variational inference or particle learning.


Comments:

- It would have been nice to compare the proposed dynamic-means algorithm with
a very simple adaptation of the DP-means algorithm: Run DP-means on one batch and
initialize means for the next batch using the results of this run. I would be
curious to see how dynamic-means compares against this alternative.

- It turns out that instead of doing small-variance asymptotics on the Gibbs sampling
updates, one could also do MAP-based asymptotics to derive hard-assignment based
variants of nonparametric Bayesian models such as DP-GMM or IBP-based latent
feature models (there is a recent paper by Broderick et al from ICML 2013:
"MAD-Bayes: MAP-based Asymptotic Derivations from Bayes"). Is it possible to derive
the dynamic-means algorithm using this approach? Please comment.

- There is work on temporal DP mixture model based on recurrent Chinese restaurant
process which is similar in spirit to dependent DP mixture models. Please see this:
"Dynamic Non-Parametric Mixture Models and The Recurrent Chinese Restaurant
Process : with Applications to Evolutionary Clustering" by Ahmad and Xing (2008).
It would be nice to have a discussion about this link of work. Please also comment
whether doing low-variance asymptotics on these methods would lead to similar
algorithms as dynamic-means?

- In the DP-means algorithm of Kulis and Jordan, the order of data points mattered.
In the cluster assignment step of the dynamic-means algorithm, does the order of data
points (within a single batch) matter? Please comment.
Summary: It is a reasonable paper, although somewhat incremental (based on the DP-means algorithm
by Kulis and Jordan). There might be some issues with its usefulness in practice (requires
choosing 3 parameters) and the experimental evidence in the paper is somewhat limited.
But I still consider the paper to be taking a step (albeit small) in an important research
direction (leveraging the flexibility of nonparametric Bayesian models for large-scale datasets).

Submitted by Assigned_Reviewer_6

The authors provide a novel small variance asymptotic analysis of the Dependent Dirichlet Process (DDP). The authors first show how to derive the small variance limits and then provide a deterministic algorithm for inference. They show that their algorithm monotonically decreases a (bounded) cost function, proving that their algorithm will converge. The authors next apply the new algorithm, Dynamic Means to synthetic data comparing to several Monte Carlo based inference methods for the DDP and the DP-Means algorithm. Finally, the authors apply their method to a real world problem of tracking aircraft trajectories, showing improved performance vs DP-Means and Gibbs sampling of the DDP.

This a reasonably interesting paper. The idea of using small variance asymptotics to derive hard clustering algorithms is not novel. However, the specific application to the DDP does raise some novel technical challenges. Deriving an algorithm which works for correlated Dirichlet processes is interesting.

The main weakness of the paper currently is the lack of adequate description of the alternative methods used for benchmarking. In particular it is not clear:

1) How hard clustering was done for the Monte Carl samplers? Was a maximum posterior estimate taken, or was a more sophisticated approach based on the posterior similarity matrices of the data points used?

2) It is not clear what particle filtering or VB algorithm the authors use.

3) The authors mention the they enforced consistency across time points when computing the accuracy metrics. Some clarification on this would help.

For the synthetic data it would be useful to show the accuracy of the methods vs time run plotted. I am extremely surprised the Dynamic Means method performs so much better than sampling based approaches, and suspect it is largely due to under-sampling for the Gibbs and PL. I realise the main point of the algorithm is that it performs much better then than sampling methods at a similar computational budget, so I think an exploration of time vs accuracy is important.

Typos:

- Line 44 "in the both the" should be "in both the"

- Line 102 $n_{k t}$ should be $n_{k \tau}$

- Lines 405-406 I am not sure why a 10 x 20 grid creates a 400 dimensonal feature vector, I would think its 200.

Minor Comments:

- It would be nice if the elements in equations (3) and (4) where in the same order.

- The text on the figures is difficult to read. A larger font would be better.
Summary: A reasonable paper which could pass the publication threshold with some revisions.
Author Feedback

Author rebuttal: The authors kindly thank the reviewers for their feedback. We have identified a few main points from the review that should be addressed: 1) The algorithm has too many free parameters, resulting in difficult tuning; 2) There is a lack of comparison with a modified DP-means algorithm; 3) Whether generalizations of the algorithm or alternate derivations are possible; 4) There is insufficient description and results in the experimental sections; and 5) Minor clarifications about aspects of the paper and the algorithm itself.

Concerning the number of parameters: A strong advantage of the present algorithm is that it possesses the combination of low computational cost and relatively smooth variation in performance with respect to its parameters. Due to these properties, automatic optimization methods are particularly applicable for parameter adjustment (e.g. gradient-based schemes[1] or Bayesian optimization[2]). There exists a large body of literature dedicated to addressing the issue of model selection[3], and as such these techniques were not mentioned in the paper; the authors will make a note of this in the final draft. Further, the existence of these parameters was not a design choice made by the authors. They are a direct result of each of the underlying processes (birth, transition, death) of the DDP mixture; any asymptotic analysis would yield a set of three parameters, as is required to describe those three actions. Finally, the authors provided a transformation to make parameter selection based on expected data characteristics more intuitive.

Regarding the proposed modified DP-means algorithm, the authors contend that the method has drawbacks which prevent devoting attention to it in a short paper. Essentially, as the cluster transition variance or death probability increases, the proposed method degrades. Two simple examples (among many) of this degradation: First, if a cluster moves further than lambda in one time step, it will always create a new cluster, and the parameter from the past time step will essentially never be used at all, preventing useful tracking; second, as the cluster death probability approaches 1, any new cluster near a recently deceased cluster will always be incorrectly linked to that deceased cluster. Correctly accounting for death and transition is what fundamentally sets the Dynamic Means algorithm apart from DP-means. In addition, the authors preferred to compare to more well-known methods, to demonstrate advantages over the state-of-the-art.

It is possible to extend this work to non-Gaussian distributions, based on recent work on small-variance asymptotics for Dirichlet process mixtures of general exponential family distributions[4]. In terms of alternative derivations, there were two approaches mentioned by the reviewers: the Recurrent CRP, and MAD-Bayes. The former does not allow for the death of clusters, and the same holds true upon taking the low-variance asymptotic limit of the Gibbs sampler. Otherwise, a similar algorithm would result from a low-variance asymptotic analysis. A MAD-Bayes-like analysis of the DDP-GMM is possible; one would wrap the transition into the parameter prior, and account for the death probability in the exchangeable partition probability function. However, based on the low-variance MAP analysis of the DP/DP-means, the low-variance MAP and Gibbs sampler analyses are expected to be similar to one another. The authors therefore decided upon the Gibbs sampler-based approach for ease of understanding.

Regarding the experimental section, the authors agree that a time budget vs. accuracy comparison would be an insightful inclusion, and will add it in the final draft. In the ADS-B section, PL and VB time results can be added into final draft. The authors understand the desire for quantitative analysis - however, L2 error isn't a good comparison metric. Indeed, comparisons of unsupervised clustering algorithms without labelled data is an open problem. Thus, the authors will hand-label a subset of the data, and provide accuracy results on that subset.

One of the reviewers was uncertain about the term "enforcing consistency" - this just means that data have to be clustered with the correct old parameter. This is of utmost importance in tracking the evolution of clusters over time. For Gibbs sampling/VB/PL, the results show reasonable clustering in each time step, but the data are being linked to the incorrect old clusters (poor tracking). Note that "enforcing consistency" is not a part of any of the algorithms; it is simply how accuracy was computed.

The comparison algorithms used in the experimental section were "PL for general mixtures" and "mean field variational inference" (as cited in the introduction). The authors agree that this was not clear in the experimental section, and citations will be added there in the final draft. Furthermore, comparisons for the sampling methods were performed conservatively (as described on lines 369-372) - for these techniques, the highest accuracy sample was selected, assuming knowledge of the true labeling. Further, the algorithms were initialized as follows (based on highest performance): Gibbs sampling was given "no initialization" (i.e. parameters and labels are introduced as necessary), and VB/PL were initialized randomly. Finally, the Dynamic Means algorithm's performance does depend on the order in which the data is assigned its labels. The authors acknowledge that this may not have been clear in the exposition of the paper, and will include that in the final draft.

1 - Yoshua Bengio. "Gradient-Based Optimization of Hyperparameters", 2000.
2 - Jasper Snoek et al. "Practical Bayesian Optimization of Machine Learning Algorithms", 2012.
3 - Joseph Kadane et al. "Methods and Criteria for Model Selection", 2004.
4 - Ke Jiang et al. "Small-Variance Asymptotics for Exponential Family Dirichlet Process Mixture Models", 2012.